# Scribble Controls Social Motivation Behavior through the Regulation of the ERK/Mnk1 Pathway

**DOI:** 10.3390/cells11101601

**Published:** 2022-05-10

**Authors:** Maïté M. Moreau, Susanna Pietropaolo, Jérôme Ezan, Benjamin J. A. Robert, Sylvain Miraux, Marlène Maître, Yoon Cho, Wim E. Crusio, Mireille Montcouquiol, Nathalie Sans

**Affiliations:** 1Univ. Bordeaux, INSERM, Neurocentre Magendie, U1215, 33077 Bordeaux, France; jerome.ezan@inserm.fr (J.E.); ben_robert@hotmail.fr (B.J.A.R.); marlene.maitre@inserm.fr (M.M.); mireille.montcouquiol@inserm.fr (M.M.); 2Univ. Bordeaux, CNRS, Aquitaine Institute for Cognitive and Integrative Neurosciences, UMR5287, 33405 Bordeaux, France; susanna.pietropaolo@u-bordeaux.fr (S.P.); yoon.cho@u-bordeaux.fr (Y.C.); wim.crusio@u-bordeaux.fr (W.E.C.); 3Univ. Bordeaux, CNRS, Centre de Résonance Magnétique des Systèmes Biologiques UMR5536, 33077 Bordeaux, France; sylvain.miraux@rmsb.u-bordeaux.fr

**Keywords:** scribble, social motivation behavior, ERK/Mnk1

## Abstract

Social behavior is a basic domain affected by several neurodevelopmental disorders, including ASD and a heterogeneous set of neuropsychiatric disorders. The *SCRIB* gene that codes for the polarity protein SCRIBBLE has been identified as a risk gene for spina bifida, the most common type of neural tube defect, found at high frequencies in autistic patients, as well as other congenital anomalies. The deletions and mutations of the 8q24.3 region encompassing *SCRIB* are also associated with multisyndromic and rare disorders. Nonetheless, the potential link between *SCRIB* and relevant social phenotypes has not been fully investigated. Hence, we show that *Scrib^crc/+^* mice, carrying a mutated version of *Scrib*, displayed reduced social motivation behavior and social habituation, while other behavioral domains were unaltered. Social deficits were associated with the upregulation of ERK phosphorylation, together with increased c-Fos activity. Importantly, the social alterations were rescued by both direct and indirect pERK inhibition. These results support a link between polarity genes, social behaviors and hippocampal functionality and suggest a role for *SCRIB* in the etiopathology of neurodevelopmental disorders. Furthermore, our data demonstrate the crucial role of the MAPK/ERK signaling pathway in underlying social motivation behavior, thus supporting its relevance as a therapeutic target.

## 1. Introduction

Social approaches, interactions and relationships rely on social memory and are built on individual abilities to differentiate other persons and remember them in order to have the appropriate behavioral responses based upon previous encounters [1]. To some extent, social dysfunctions are characteristic symptoms of various psychiatric illnesses, including several neurodevelopmental disorders (NDDs) such as autism spectrum disorder (ASD), but are also found in mood and anxiety disorders with roots in prenatal life [2]. Given the importance of social interactions in our everyday life, it is crucial to understand their bases at the molecular, neural and network levels to develop new pharmacotherapeutics that can ease dysfunctional social behaviors.

The *SCRIB* gene (SCRIBBLE, SCRB1; OMIM# 607733) encodes for a protein named Scribble (Scrib), which is key in many developmental processes including cell proliferation, migration and polarity [3]. Scrib is known for its involvement in planar cell polarity signaling that regulates embryonic and postnatal development [4,5,6,7]. Mutations in SCRIB are described in patients with spina bifida [8,9,10], one of the most common forms of neural tube defect that is found at a high frequency in ASD patients [11,12,13]. Accumulating evidence suggests a role of SCRIB, together with other polarity genes, in neurodevelopmental disorders (for a review, see [14]). Mutations in the locus encompassing the SCRIB gene (8q24.3) have been linked to the rare Verheij syndrome (VRJS, OMIM #615583), characterized by severe growth retardation including microcephaly and intellectual disability [15], and ASD [16,17,18]. Scrib also modulates MAPK signaling and interacts directly with ERK [19], and chromosomal deletions encompassing the ERK-encoding *MAPK* genes are commonly associated with NDDs and ASD [20,21,22]. Furthermore, abnormalities in ERK functionality have been described in several mouse models of NDDs and ASD [2,23,24,25]. In our previous work, we have also shown a major role of Scrib in modulating dendritic arborization and connectivity, post-endocytic NMDA receptor trafficking and hippocampal synaptic plasticity [5,26] (i.e., processes that are known to play a key role in the etiopathology in NDDs and ASD) [23,27,28].

The potential link between *SCRIB*, brain development and function and underlying ASD-like behaviors is poorly understood. Additionally, understanding the molecular mechanisms involved in the neurodevelopmental pathologies associated with *SCRIB*, including autism, forms the basis of ongoing work to identify new biomarkers for autism [29]. Here, we conducted an extensive neurobehavioral characterization of *Scrib^crc/+^* mice to investigate a link between this gene and ASD-like behaviors. We have chosen to focus our study on social behaviors, as they represent the core symptoms of ASD that are also common to NDDs. We used a variety of tests to evaluate whether the potential impact of Scrib mutation on social behaviors could (1) be confounded by the interference of other behavioral alterations (e.g., changes in general activity, emotionality or olfaction) or (2) be specific to this domain or instead affect the general novelty preference and detection (e.g., object recognition, spontaneous alternation and olfactory habituation). We correlated these behavioral results with MRI and c-Fos expression analysis, as well as an exploration of the role of the MAPK/ERK pathway. Mnk1 kinase is a known downstream target of ERK signaling, and a recent study suggested it could be a part of a molecular signature for ASD [30,31]. We were able to rescue these behavioral deficits by systemic administration of MEK and Mnk1 inhibitors of the MAPK/ERK pathway in adult mice. This supports the hypothesis that the MAPK/ERK pathway is hyperactive in *Scrib^crc/+^* mice and that it underlies sociability deficits. Altogether, our results suggest that social behaviors are controlled by Scrib regulation of the ERK-Mnk1-overlapping mechanisms of phosphorylation.

## 2. Materials and Methods

### 2.1. Animals

The genetic background of the transgenic animals line was 50% NMRI, 25% BALB/C and 25% C57BL/6 [32,33]. The mutant line was maintained by random breeding between heterozygous *Scrib^crc/+^* and wild-type *Scrib^+/+^* mice. The mice carrying the Crc mutation reproduced normally and were maintained as described previously [33]. Experiments were performed using heterozygous mice for the mutation (*Scrib^crc/+^*) and WT (*Scrib^+/+^*) littermates. Several cohorts of animals and multiple behavioral tests were used. Whenever possible, naïve animals were employed for behavioral testing. When the same cohort was used for multiple tests, the most stressful assays were administered last to minimize between-test interference.

### 2.2. Behavioral Analyses

For our behavioural analyses, we used sexually mature mice (in particular for the social reward test) that we started handling at around 8 weeks, and then we performed all our experiments on adult animals from the age of 10–11 weeks. All subjects were housed in collective cages under standard laboratory conditions with a 12-h light and 12-h dark cycle (light on: 7:00 a.m.) with food and water supplied *ad libitum*. To avoid any experimental bias, we used these naive animals of non-mutant lineage as the social stimulus. Juvenile male Swiss mice (21–28 days old, Janvier, France) or SV27 female mice (10–12 weeks) were used for social interaction and communication analysis that were not raised in the same animal house and that had never been in contact with the crc lineage or animals with social deficits. All animals were assigned randomly to the different experimental conditions. One group (*n* = 8 WT and *n* = 16 *Scrib^crc/+^*) was tested in an open field and in a light-dark test. The second group (*n* = 13 WT and *n* = 12 *Scrib^crc/+^*) was tested in a plus maze, Y-maze, pre-pulse inhibition and social communication. Four naive groups were used for social interest and preference for social novelty in the three-compartment test and olfactory tests (*n* = 9 WT and *n* = 10 *Scrib^crc/+^*), the preference for social reward (*n* = 7 WT and *n* = 11 *Scrib^crc/+^*), social habituation (*n* = 11 WT and *n* = 15 *Scrib^crc/+^*) and novel object recognition (*n* = 10 WT and *n* = 10 *Scrib^crc/+^*). Different groups (*n* = 8–10 WT and *n* = 7–10 *Scrib^crc/+^*) were used for the ERK and Mnk1 inhibitor experiments. One last group was used for the self-grooming and marble tests (*n* = 9 WT and *n* = 10 *Scrib^crc/+^*). All tests were performed during the light phase of the light-dark cycle. All experimental apparatuses were cleaned with 70% ethanol or Phagospray DM between subjects to remove residual odors. For the other experiments, *n* = 3–4 WT and *n* = 5–6 *Scrib^crc/+^* were used for c-Fos analysis, while *n* = 3 WT and *n* = 3 *Scrib1^crc/+^* was used for western blot, IRM and 3D analysis.

### 2.3. Assessment of Social Behaviors

The multiple social tests used assessed multiple characteristics of social behaviors in different social contexts. All data from the three-compartment test were analyzed with Ethovision (Version 13, Noldus).

### 2.4. Social Motivation Task in the Three-Compartment Test

The three-compartment test provided an evaluation of a social preference without direct social interaction, avoiding inter-male aggression or sexual behaviors. The test was performed as previously described [5]. It consisted of 3 10-min trials. During trial #1 (habituation), the tested mouse was allowed to explore the 3 chambers, in which each end chamber contained an empty small wire cage. In trial #2 (social interest), a stranger male mouse (S1, social stimulus) was placed under a wire cage in one end chamber while an object (O, nonsocial stimulus) was placed in the opposite end chamber. For trial #3 (social novelty), a second stranger male mice (S2) replaced the object. The tested mouse was free to choose between a caged novel stranger (S2, novel social stimulus) versus the same caged mouse in trial 2 (S1, familiar social stimulus).

### 2.5. Preference for a Social Reward in the Three-Compartment Test

The mouse was habituated to the 3-compartment apparatus for 10 min in the empty compartment. Then, we tested the social reward over 5 min. We placed a novel adult male stranger mouse (M) under a cage in one end chamber, and an adult female stranger mouse (F) was placed in the opposite end chamber (adapted from Pietropaolo et al., 2011 [34]). The tested mouse was free to choose between exploring one of the stimuli (F or M). Exploration was assessed by automatically measuring the time spent in each contact area containing the stimulus cage during 5 min.

### 2.6. Direct Social Interaction and Communication

The quality of the behavioral patterns of the social behaviors were assessed using the direct social interaction test with an adult female in in the home cage [34]. This experimental setting is also typically used to analyze ultrasonic vocalizations in adult male mice [35]. The tested mice were isolated 24 h before the test. An unfamiliar stimulus mouse (adult NMRI female) was introduced into the testing cage and left for 5 min. Testing sessions were recorded and videos were analyzed with Observer XT (Noldus), taking only the tested mice into account. An observer (blind to the animal genotype) scored the time spent performing each of the following social behaviors: sniffing the head and the snout of the partner, its anogenital region or any other part of the body. During the test, an ultrasonic microphone (CM16/CMPA, Avisoft, Berlin, Germany) was suspended 2 cm above the cage lid. Vocalizations were recorded and analyzed afterward as previously described [34].

### 2.7. Social Habituation

Social habituation was analyzed through repeated encounters with a juvenile male to elicit social interest but minimize the risk of aggression [34]. The test was performed on two consecutive days. On day 1, an unfamiliar stimulus mouse (a juvenile Swiss male, 3–4 weeks old) was introduced in the home cage of the tested subject and left for 2 min. This procedure was repeated on day 2 with the same stimulus mouse. Social investigation of the stimulus mouse by the tested subject was scored by a trained observer, who timed the duration of the investigation with a handheld stopwatch. Behaviors that were scored as social investigation included direct contact, sniffings and close following (<1 cm).

### 2.8. Open Field Activity and Nonsocial Neophobia

Open field activity was assessed during 30 min in brightly lit enclosures (40 × 40 cm) equipped with infrared photocells to measure horizontal movements. Then, a novel object (a cup 7.5 cm in height and 6.5 cm in diameter) was placed upside-down into the center of each open field, and the mouse behavior was tested for an additional 30 min [36]. The percentage of entries and time spent in the center (15 cm in diameter), as well as the overall motor activity, were quantified automatically by software (Videotrack, Lyon, France).

### 2.9. Elevated Plus Maze

The plus maze was made of gray Plexiglas and was elevated approximately 60 cm from the floor. It had two open and two enclosed arms radiating outward from a central open square. The mice were placed in the center of the maze and allowed free access to all arms for 5 min [36]. The maze was illuminated by a halogen light at 100 lux in the central open square. After each assessment, the equipment was cleansed with Phagospray DM to remove residual odors. Tracking images from a camera above the center of the apparatus were analyzed with Ethovision (Version 13, Noldus Technology, Wageningen, The Netherlands). The percentage of time spent in the open arms was computed.

### 2.10. Dark-Light Emergence Test

The emergence test [37] was conducted in an open field (MEASURES) containing a polypropylene cylinder (10 cm deep and 6.5 cm in diameter) located lengthwise along the wall, with the opening located at a distance of 10 cm from the corner of the open field. Each subject was placed in the cylinder and left undisturbed in the apparatus for 15 min. The latency to leave the cylinder, defined by exiting with all four paws in the open field, was measured, as well as the number of entries and the time spent in the cylinder. The overall motor activity was quantified as the total distance performed during the test. The equipment was cleansed with Phagospray DM between animals to remove residual odors.

### 2.11. Self-Grooming Test

The mice were scored for spontaneous grooming behaviors. A WT or *Scrib^crc/+^* mouse was placed individually in a new standard mouse cage with fresh bedding (46 cm long × 23.5 cm wide × 20 cm high, illuminated at 50 lux). After a 10-min habituation period in the test cage, each mouse was scored with a stopwatch for 10 min for the cumulative time spent grooming all body regions [36].

### 2.12. Marble-Burying Test

This test is considered an indicative measure of repetitive behavior in a novel environment related to digging behavior. The test consists of placing 20 glass marbles, spaced in 5 rows of 4 on an approximately 5-cm layer of sawdust bedding, in a plastic cage. Mice are individually placed in the cage, and after 30 min of testing, the number of marbles buried with sawdust is counted. The marbles are considered buried if they are at least one half covered with bedding [36].

### 2.13. Buried Food Test

On the day of the test, the mice were placed in a new cage (46 × 23.5 × 20 cm) containing 3 cm of bedding and allowed to explore for 10 min (adapted from Yang and Crawley 2009 [38]). The animal was removed, and a piece of cheese was buried under 2 cm of bedding. The time required to find the food was measured by an observer.

### 2.14. Olfactory Habituation and Dishabituation

For the test of olfactory habituation and dishabituation (adapted from Yang and Crawley, 2009 [38]), different odors (two nonsocial odors and two social odors) were presented in consecutive trials of 1 min each. First, the mouse was placed in the testing cage for habituation with a cotton applicator placed through a hole in the grid cage. After 30 min, a 10-µL drop of orange extract (Sanoflore, Gigors-et-Lozeron, France; 10^−6^ dilution) was placed on the cotton tip part of the applicator in the home cage for 1 min. The presentation of this odor stimulus was repeated 3 times with 10-min intervals. In a fourth trial 10 min later, we placed a new applicator containing a 10-µL drop of lavandin extract (Sanoflore, France; 10^−6^ dilution) in the subject’s cage for 1 min. Ten minutes later, a piece of white paper containing male or female urine (a mixture of urine from 10 unfamiliar adult mice) was placed in the subject’s cage sequentially in 4 consecutive trials of 1 min each. We considered olfactory investigation as the time sniffing the applicator.

### 2.15. Prepulse Inhibition of the Acoustic Startle Reflex

The apparatus (SR-LAB, San Diego Instruments, San Diego, CA, USA) and procedures were previously described in detail in [39,40,41,42]. Briefly, the animals were acclimated to the apparatus for 5 min. The first 6 trials consisted of 6 pulse-alone trials, with 2 for each pulse intensity (100, 110, or 120 dBA), presented in a pseudorandom order. Subsequently, 10 blocks of trials were presented. Each block consisted of 3 pulse-alone trials, with 1 for each pulse intensity, 3 prepulse-alone trials (+6, +12, or +18 dB units above the background of 65 dBA), 9 possible combinations of prepulse-plus-pulse trials (3 levels of pulse × 3 levels of prepulse), and 1 no-stimulus trial (i.e., background alone). These 16 trials were presented in a pseudorandom order within each block, with a variable intertrial interval of a mean duration of 15 sec. The session was concluded with a final block of six consecutive pulse-alone trials as in the first block. The reactivity scores obtained for the first and the last blocks of six consecutive pulse-alone trials were separately analyzed to measure the startle habituation. The data obtained in the remaining trials were categorized into three main different subsets. First, the startle reactivity was assessed by the reactivity scores obtained in the intermediate pulse-alone trials. Second, the reactivity for the prepulse-plus-pulse trials relative to the middle pulse-alone trials was used to estimate the prepulse inhibition. PPI was analyzed by converting the reactivity data into percent scores (%PPI = 100 × (pulse-alone − prepulse-plus-pulse)/pulse-alone) calculated for each subject for each of the nine possible prepulse-plus-pulse combinations. Third, to measure the prepulse-elicited reactivity, we included the data from the prepulse-alone and no-stimulus trials.

### 2.16. Recognition of Nonsocial Novelty

Each subject was placed in the center of an open field identical to the one previously described and allowed to explore for 10 min (habituation phase). After habituation, two identical objects (orange china cup 7.5 cm in height and 6.5 cm in diameter) were placed in two opposite corners of the open field, and the experimental animal was left in the apparatus for 5 min (sample phase). At the end of this session, the mouse was returned to his or her home cage. One hour later, one of the two objects was replaced by a novel object once (gray plastic rectangle object: 4 × 8 cm), and the animal was given a 5-min exploration session (test phase) (adapted from Leger et al., 2013 [43]). The time spent exploring each object (sniffing or staying within a 1-cm distance) was measured during both the sample and test phases by an observer.

### 2.17. Spontaneous Alteration in the Y-Maze

Spontaneous alternation was assessed in a gray plastic Y-maze placed on a table 80 cm high and located in the middle of a room containing a variety of extra maze cues. The three arms of the Y-maze were similar in appearance and spaced 120° from each other. Each arm was 42 cm long and 8 cm wide. The entire maze was enclosed by a wall 15 cm high and 0.5 cm thick. The mice were introduced at the end of one of the arms and allowed to explore the maze for 5 min. Allocation of the start arm was counterbalanced within the experimental groups. An entry into one of the arms was scored by an observer unaware of the genotype of the animals when all four paws of the animal were placed inside an arm. Spontaneous alternation, expressed as a percentage, refers to that proportion of arm choices differing from the previous two [44,45]. Thus, if an animal made the following sequence of arm choices—A, B, C, B, A, B, C, and A—the total number of alternation opportunities would be 6 (total entries minus 2), and the percentage alternation would be 67% (4 out of 6).

### 2.18. Treatments

The MEK inhibitor α-[amino[(4-aminophenyl)thio]methylene]-2-(trifluoromethyl)benzeneacetonitrile (SL327) (Sigma-Aldrich, St. Louis, MO, USA) was dissolved in a vehicle solution of 2.5% dimethyl sulfoxide (DMSO) and 2.5% Cremophor EL saline solution at 30 mg/kg [46]. The Mnk1 inhibitor CGP57380 (Sigma-Aldrich) was dissolved in a vehicle solution of 2.5% dimethyl sulfoxide (DMSO) at 10 mg/Kg [30]. SL327 and CGP57380 were injected intraperitoneally (IP) at a volume of 3.3 mL/kg 1 h before the social interest test. The control mice received the same volume of the vehicle.

### 2.19. Analysis Following the Social Interest Test

For each genotype, we analyzed two groups of mice: one group exposed to the social interest test (habituation and social interest = social group) and one group unexposed to the social interest test (habituation alone = control group). The social group went through the two trials of the three-compartment test described previously (trial 1 + trial 2). During the same period, the control group was exposed to the habituation test (trial 1) during the same time in the same experimental room. At the end of the test, all mice were returned in their home cage and left undisturbed for 60 min.

### 2.20. c-Fos and Zif-268 Immunoreactivity

The mice were anesthetized using Phenobarbital i.p injection and transcardially perfused with physiological saline for 5 min followed by 4% paraformaldehyde diluted in 0.1 M phosphate buffer with a pH of 7.4 (PFA; 4%) for 15 min. The brain was removed, postfixed for 24 h in 4% PFA, and cut with a vibratome. The free-floating coronal sections (40 µm) were incubated in 0.3% Triton X-100, 0.3% normal goat serum, and c-fos or Zif-268 primary antibody (1/1000 and 1/500; Santa Cruz Biotechnology, Dallas, TX, USA). Next, the sections were incubated with biotinylated goat anti-rabbit IgG (Vector Laboratories, Burlingame, CA, USA) followed by an ABC EliteKit (Vector Laboratories). The immunoreactive cells were visualized using a diaminobenzidine (DAB, Vector Laboratories) colorimetric reaction. Images were acquired with a Leica microscope, and the number of c-Fos- and Zif-268-positive cells was quantified automatically using Metamorph. The brain areas were identified using the stereotaxic atlas of Paxinos and Franklin (1997) [47].

### 2.21. Laser Capture Microdissection (LCM) Analysis

The mice were anesthetized by brief inhalation of isoflurane (5% in air), sacrificed by decapitation, and the brain was rapidly dissected, snap frozen, and stored at −80 °C. LCM was performed on the coronal frozen sections as described in [48]. The anterior hippocampal neuronal layers (B = −0.94 at −2.7 mm), including the CA3, the CA1, and the DG, were selectively captured on three different LCM caps. Following LCM, 150 µL of extraction buffer was added to the caps and stored at −80 °C until protein isolation.

Western blot analysis. Preparation for the protein extracts was performed as previously described in [5]. The concentration of protein was measured by a BCA assay kit, and western blots were performed on 10% SDS-PAGE using standard methods [49]. The antibodies used were anti-Scrib antibody (1/500; AbMM468 [7]), anti-p44/42 MAPK (1/10,000; #9102), and anti-Phospho-p44/42 MAPK (1/1000; #9101) from Cell Signaling (Danvers, MA, USA). The secondary antibody used was anti-rabbit HRP-conjugated from GE Healthcare (Chicago, IL, USA).

### 2.22. Cell Culture, Transfection, and the SRE-Luciferase Reporter Assay

The HEK293T cells were maintained in Dulbecco’s Modified Eagle’s Medium (GIBCO, Waltham, MA, USA) supplemented with 10% fetal bovine serum (GIBCO) and were transfected using linear polyethylenimine (MW 25,000; Polysciences, Warrington, PA, USA). An SRE-Luciferase reporter assay was performed using the Dual-Luciferase assay system (Promega, Madison, WI, USA) as described previously [50]. Each well in the 12-well plates received 0.05 µg of pTKRenilla-Luc, 1 µg of pGL4.33[*luc2P*/SRE/Hygro] Vector DNA (Promega), and indicated amounts of human Scrib constructs (pCDNA3-eGFP-Scrib, pCDNA3-eGFP-Scrib^Crc^). The total DNA was adjusted to 2.05 µg by supplementing pCDNA3 DNA. One day after transfection, the cells were serum starved for 2 h and then treated with 20% FBS containing the medium for an additional 6 h before being harvested in 200 µL of 1X passive lysis buffer for luciferase activity measurement. The results are expressed in terms of relative luciferase activity (ratio of firefly luciferase activity divided by Renilla luciferase activity), which was determined using a POLARstar Omega multi-mode microplate reader (BMG LABTECH, Ortenberg, Germany).

### 2.23. Magnetic Resonance Imaging (MRI)

The experiments were performed on a 4.7 T Bruker system (Ettlingen, Germany) equipped with a 12-cm gradient system capable of a 660 mT/m maximum strength. Measurements were performed with a birdcage resonator (25 mm in diameter and 30 mm long) tuned to 200 MHz. The mice were anesthetized with isoflurane (1–1.5% in air) and maintained at a constant respiration rate of 75 ± 15 respirations/min. The animals were placed in a lying position within the magnet with the head at the center of the NMR coil. A 3D TrueFISP imaging sequence with an alternating RF phase pulse method and sum of square reconstruction were used as already described in [51,52]. The following parameters were used: TE/TR: 2.5/5 ms; flip angle: 30°; bandwidth: 271 Hz/pixel; FOV: 20 × 20 × 16 mm; matrix: 128 × 128 × 102; spatial resolution: 156 × 156 × 156 mm^3^; number of averages: 4; 4 DeltaPhi values; and total acquisition time: 17 min 4 s.

### 2.24. D Volume Reconstitution and Surface Rendering

Consecutive z-series of coronal brain sections were stained with cresyl violet and scanned with Hamamatsu NANOZOOMER 2.0 (Hamamatsu, Japan). All brain images were compiled into a stack file after being aligned using ImageJ software. Then, 3D reconstruction and volume measurement were performed with Imaris Scientific 3D/4D image processing and analysis software.

### 2.25. Statistical Analysis

Normality was tested with the Shapiro–Wilk normality test. Statistical analysis was performed using an unpaired Student’s *t*-test and Mann–Whitney, one-way, two-way, or three-way ANOVA analysis, followed Bonferroni’s for ANOVA. The analysis was performed using GraphPad Prism 8 software. Effects with *p* ≤ 0.05 were considered statistically significant.

## 3. Results

### 3.1. Scrib^crc/+^ Mice Had Deficits in Social Motivation Task (or the Three-Compartment Test for Social Preference)

We tested social interest and the preference for a social novelty paradigm using the three-compartment test (Figure 1A). In the social interest test (Figure 1C), the wild-type (WT) mice had a stronger preference for the stranger male mouse (S1, social stimulus) than the nonsocial novel stimulus (O), whereas the *Scrib^crc/+^* mice showed no preference for the stranger male. In social novelty preference (Figure 1D), the WT mice showed a preference for the novel social stimulus (S2) over the familiar social stimulus (S1), while this preference was not observed in the *Scrib^crc/+^* mice (Figure 1D). For all trials, the number of entries in both stimulus chambers was similar for both genotypes (Appendix A), suggesting no difference in the general exploration levels.

We then assessed the preference for a more attractive, rewarding stimulus (i.e., an adult female (F), in the three-compartment apparatus, compared to a stranger male mouse stimulus (M)) (Figure 1B). During the test, the WT mice showed a preference for the female stimulus (F), which was absent in the mutants (Figure 1E). It should be noted that in the direct interaction test with the female, the number of interactions generated was quite comparable to the controls (Appendix A). These data mean that the *Scrib^crc/+^* mice had a clear deficit in the natural preference for social novelty and social reward without changes in general direct social investigation.

### 3.2. Scrib^crc/+^ Mice Displayed No Deficits in Direct Social Interaction and Communication

Next, we evaluated the ultrasonic vocalizations during direct social behavior. We found no difference in the time spent in social investigation or in the ultrasonic vocalizations emitted during the social interaction session between genotypes (Figure 1F and Appendix A), indicating that the *Scrib^crc/+^* mice had no major defect in direct social interaction and no communication defects with an adult female. In a social habituation paradigm (Figure 1G) that provided a measure of hippocampus- and amygdala-dependent social memory [53], the WT and *Scrib^crc/+^* mice showed characteristic social habituation (i.e., a decrease in the duration of social investigation with a juvenile male), but the *Scrib^crc/+^* mice presented slower habituation compared with the WT mice (Figure 1G), indicating that the *Scrib^crc/+^* mice had a deficit in social habituation. This decrease was not due to a general deficit in olfactory investigation because the presentation of successive different odors resulted in a similar amount for the control group (see Appendix A). Therefore, it was concluded that the *Scrib1^crc/+^* mice with persistent interest during repeated presentations failed to develop social memory.

In summary, in the direct interaction test, physical contacts were numerous, active, and reciprocal for both the control and *Scrib^crc/+^* mice. It should be noted that the *Scrib^crc/+^* mice reproduced normally. However, when the mice were subjected to a social choice, during the social motivation task, there was a clear deficit in social preference and recognition. It is therefore this last very specific behavioral deficit that we want to study in more detail.

The social deficits in *Scrib^crc/+^* mice are not a consequence of anxiety, alterations in locomotion, neophobia, or olfactory deficits, which are all crucial for social interest and novelty (Appendix A). They are also not associated with the problem of stereotyped behaviors or sensory-motor integration (Appendix A). Altogether, our data show that the motor function, general novelty recognition problems, or sensory inputs tested were normal in the *Scrib^crc/+^* mice.

### 3.3. c-Fos Levels Were Increased in the DG and CA3 Regions of the Hippocampus in Scrib^crc/+^ Mice after Social Exposure

We next mapped the brain regions activated or not activated during the social interest test in the *Scrib^crc/+^* and WT mice (Figure 2). After the test, all mice were left undisturbed in their home cages for 1 h before perfusion (social condition). In our control group, both the WT and *Scrib^crc/+^* mice were placed in the empty chamber for the same time and then returned to their home cages until tissue collection (control condition). After the social test, in WT mice, we observed an increase in c-Fos-positive cells compared with the control condition in restricted areas of the basal forebrain, including the hippocampus areas (DG, CA1, and CA3), enthorinal cortex (EntCx), medial nucleus of the amygdala (MeA), motor (MCx) and piriform cortices (PirCx), and striatum (ST), as well as in the granular cells of the accessory olfactory bulb (AOB Gcells) (Appendix A). In the *Scrib^crc/+^* mice, the same regions were activated, but the density of the c-Fos-positive cells was also significantly increased in the sensorimotor cortex (S1Cx) and in the cortical nucleus of the amygdala (CoA). In addition, the dorsolateral part of the striatum (STDL) was not activated in the *Scrib^crc/+^* mutant compared with the controls (Figure 2B and Appendix A). In both the control and social interest condition, this pattern was modified by the mutation. The number of c-Fos-positive cells was significantly higher in the mitral cells of the olfactory bulb (AOB/MOB) and lower in the piriform cortex (PirCx) in the *Scrib^crc/+^* mutant compared with the WT mice (Figure 2B and Appendix A).

Thus, in these structures, the Scrib expression levels affected c-Fos activity independently from the social behavior task. However, after the social interest test, a more robust increase in the number of c-Fos-positive cells was observed in the CoA (≈+56%), DG (≈+12%), and CA3 (≈+55%) regions of the hippocampus and in the S1 region of the cortex (≈+48,7%) in the *Scrib^crc/+^* mice compared with the WT mice but not in the CA1 or the medial nucleus of the MeA (Figure 2D and Appendix A). This solid upregulation of the c-Fos levels in the CA3 and DG of the *Scrib^crc/+^* mice was a direct consequence of the social interest exposure, because the expression levels of c-Fos in the control animals were comparable between the WT and *Scrib^crc/+^* mice in these two regions (Figure 2D). In the CA3 and DG of the hippocampus, we found a similar upregulation of *Zif268*-positive cells (Appendix A). Together, these results show that the DG and CA3 regions of the hippocampus were the main activated ones during the social motivation test in the *Scrib^crc/+^* mutant mice.

### 3.4. Reduced Structural Integrity of the Hippocampus in Scrib^crc/+^ Mice

Magnetic Resonance Imaging (MRI) analysis showed no major changes in the global volume of the brain (Figure 3), confirming a previous study [5], but we observed an increase in the volume of the olfactory bulbs and a decrease in the volume of the hippocampus (Figure 3A,B), which was confirmed by 3D reconstruction analysis of the histological serial sections (Figure 3C,D). Interestingly, in our previous paper [5], we did not find any difference in the hippocampus when using classical 2D analysis on a few tissue sections. By combining in vivo MRI analysis and a 3D rendering of the serial fixed sections in adult mice, we showed an anatomical alteration phenotype in the hippocampus for the *Scrib^crc/+^* mice.

### 3.5. Reduction of Scrib Levels Leading to a Significant Increase in ERK Phosphorylation in the Hippocampi of Scrib^crc/+^ Mice after Social Exposure

Scrib inhibits the activation of extracellular signal-regulated kinase (ERK) pathways in various systems [3,19,54], and ERK and the mitogen-activated protein kinase (MAPK) signaling pathway are linked to neurodevelopmental disorders, including ASD [2,23,24,25]. Our data show that ERK1 and ERK2 were activated in the three regions of the hippocampus for both the *Scrib^crc/+^* and WT mice, which performed the social interest test (Figure 4A–C). The Scrib reduction levels in the *Scrib^crc/+^* mice led to a hyperactivation of ERK during social exposure in the DG and the CA3 but not in the CA1 compared with the WT mice.

### 3.6. The Scrib^crc^ Form Was Crucial for ERK Pathway Activation

We used a classical ERK signaling pathway in vitro reporter assay using SRE-LUC as a reporter gene to evaluate the Scrib and Scrib*^crc^* impact on ERK activation. As expected, (Figure 4D), full-length Scrib reduced the serum-induced SRE-LUC reporter gene expression in a dose-dependent manner. The Scrib*^crc^* truncated form of Scrib exhibited a biphasic response. Lower doses significantly increased the luciferase expression when compared with full-length Scrib, while higher doses were inhibitory, similar to full-length Scrib, although not to the same extent. These results are consistent with an inhibitory role of Scrib on the ERK pathway, inhibition that was reduced in the Scrib*^crc^* mutants most probably because of the absence of an ERK binding site in the truncated protein. The mechanisms leading to the potentiating effect observed for lower levels of Scrib*^crc^* are unclear at the moment, but it could be the result of a dominant negative effect by Scrib*^crc^* on full-length Scrib. This dual effect is consistent with ERK hyperphosphorylation in the *Scrib^crc/+^* mice and could explain part of the complex phenotype in these mice.

### 3.7. Specific Inhibition of ERK Rescued the Social Motivation Deficits in the Scrib^crc/+^ Mice

We attempted to rescue the lack of social interest in the *Scrib^crc/+^* mice by inhibiting ERK overactivation in the hippocampus with an inhibitor of the mitogen-activated protein kinase (MEK) SL327 [55]. The *Scrib^crc/+^* mice treated with 30 mg/kg SL327 showed normal levels of social interest comparable to their saline-treated WT littermates (Figure 5A,D). The effect was comparable to the WT in the saline condition and different from the *Scrib^crc/+^* mice in the saline condition (Figure 5D). The same rescued effects were observed in the preference for social novelty (Figure 5E).

### 3.8. Downstream of ERK Signaling, the Specific Inhibition of Mnk1 in Scrib^crc/+^ Mice Also Rescued Social Motivation Deficits

To determine the participation of Mnk1, a downstream target of ERK signaling suggested to be a molecular signature for ASD [23,30], in the social deficits of the *Scrib^crc/+^* mice, we used CGP57380, a cell-permeable pyrazolo-pyrimidine compound (Figure 5A). Mnk1 inhibition improved the social interest deficits in the *Scrib^crc/+^* mice (Figure 5D), as well as the defects in preference for social novelty (Figure 5E). The impairments in social recognition observed in the *Scrib^crc/+^* mutant were thus due, at least in part, to an increase in ERK/Mnk1 pathway activity. Furthermore, we demonstrated in our mouse model that a social interest deficit can be rescued in adult animals by modulating two biological targets which are part of a major signaling pathway.

## 4. Discussion

This study provides evidence for the role of Scrib in the regulation of social preference and recognition, mediated by hippocampal functionality and more specifically by the ERK/Mnk signaling pathway. First, we report that the *Scrib^crc/+^* mice had behavioral alterations that were specific to the social preference and recognition domain in the absence of changes in other abilities. Second, these social deficits are associated with increased c-Fos activity in the hippocampus and an overactivation of phosphorylated ERK. Third, the social abnormalities of the *Scrib^crc/+^* mice were rescued by the administration of MEK, ERK, and Mnk1 inhibitors. Our study identifies that a hyperactivation of ERK and Mnk and a dysregulation of the ERK/Mnk1 signaling pathway in the absence of a fully functional Scrib protein are key molecular mechanisms that trigger social behavior defects in the *Scrib^crc/+^* mutant mice. Taken together, our findings suggest that the control of ERK and Mnk1 phosphorylation by Scrib-dependent mechanisms is critical for social interest and recognition.

The *Scrib^crc/+^* mice displayed a lack of social interest and preference for social novelty and social reward, as well as reduced social habituation. The qualitative patterns of social interaction, as assessed in the direct social interaction test, were unaltered, as well as ultrasonic communication. As we performed mainly quantitative measures of the ultrasonic frequency and duration, it is still possible that subtle differences in communication exist, such as through qualitative spectrographic analyses, as suggested by studies on mouse models of ASD [56]. Nonetheless, our findings indicate that social recognition and the detection of social novelty represent the major domains that are robustly affected by *Scrib^crc^* mutation. Importantly, the social deficits of the *Scrib^crc/+^* mouse mutant were not accompanied by other socially related behavioral alterations. This suggests that the role of *SCRIB* in the etiopathology of ASD would be, in part, specific for social alterations without other additional sensory-motor or emotional symptoms or cognitive deficits. We have shown that the *SCRIB* mutation can improve some cognitive abilities, such as higher spatial memory in the water maze task [5]. This type of improvement is also observed in some patients with ASD that are high-functioning or have Asperger’s syndrome or mice mutants for the ADS-related gene (NL3 KI/Shank1 KO), and it will be interesting to conduct a further study on this point. Nevertheless, the social phenotype represents the most relevant core symptom of ASD-related pathologies, and it is common to a variety of NDDs. The *Scrib^crc/+^* mice could represent a good model to study specific types of ASD or NDDs. Interestingly, other polarity genes have been linked to NDDs and some more specifically to ASD-like phenotypes [14].

We found no major structural alterations in the brains of the mutants except for larger olfactory bulbs and a smaller hippocampus. MRI studies have highlighted a reduced volume of the amygdala or hippocampus in children and adult ASD patients [57,58] or in adult patients with spina bifida [59]. It has recently been shown that a brain network composed of the hippocampus, mPFC, ACC, and amygdala is required for the consolidation of social recognition memory [60]. In the *Scrib^crc/+^* mice, both the hippocampus and amygdala showed an excessive c-Fos induction, a result that might highlight an abnormal regulation of neuronal activity, resulting in deficient social memory. Our data are therefore in agreement with studies suggesting that the hippocampus plays a critical role in social memory and recognition [53,61,62,63,64]. It will be interesting to further explore the role of the hippocampus in these different forms of memory.

Though various signaling pathways have been associated with ASD, MAPK signaling is linked to neurodevelopmental disorders and, more specifically, ASD [2,24,25,61]. Because the Scrib levels are reduced in *Scrib^crc/+^* mice, we believe that the observed hyper-phosphorylation of ERK1 and 2 in the mutant was the result of a lack of inhibition by the MAPK cascade. The biphasic regulation of the mutated *Scrib^crc^* form of Scrib suggests a tight and dynamic regulation of MAPK activity depending on the levels of Scrib but also on the domains of the proteins available in the neurons. Such a regulation could be region-specific. A similar complex effect of a mutation can be observed in a neuroligin-3 knock-in mouse, with a R451C-substitution likely acting as a gain-of-function mutation [65]. In this study, the authors observed that although only 10% of the neuroligin-3 protein remained in the knock-in mice, this remaining neuroligin-3 protein lead to an inhibition of synaptic transmission, whereas the full neuroligin-3 KO mice exerted no such effect. Consistent with this, chromosomal duplications or deletions of 16p11.2, which includes the MAPK3 gene encoding for ERK1, or a microdeletion on chromosome 22, which contains the MAPK1 gene coding for ERK2, are commonly found to be associated with neurodevelopmental disorders and ASD [20,21,22].

Mnk has emerged over the years as an important target to alleviate some ASD-related issues [66]. A recent study further suggests that MAPK activity might constitute a molecular signature of clinical severity in autism [31]. In that study, the authors suggested that the cascade ERK1-2/Mnk1/eIF4E may constitute a molecular signature predictive for the early diagnosis of autism, notably severe ASD. Our results showed that inhibiting Mnk1 has similar effects to directly inhibiting ERK phosphorylation, both for sociability and social novelty behavior, strongly suggesting it is a similar cascade that is deregulated in *Scrib^crc/+^* mice.

## 5. Conclusions

Altogether, this work identifies Scrib as an important regulator of select social behaviors in mice through negative regulation of the ERK/Mnk signaling pathway and that the *circletail* mutant mouse is a useful model for which the effects on social preference and the recognition of candidate therapeutical compounds can be tested. It is tempting to suggest that the disruption of SCRIB-dependent signaling could underlie common genetic factors to NTD and ASD-like deficits and may, in some way, predispose the development of these pathologies. In the future, it will be important to define the upstream and other downstream effectors of Scrib implicated in the processing of the sensory inputs that play a pivotal role in social information integration.

## Figures and Tables

**Figure 1 cells-11-01601-f001:**
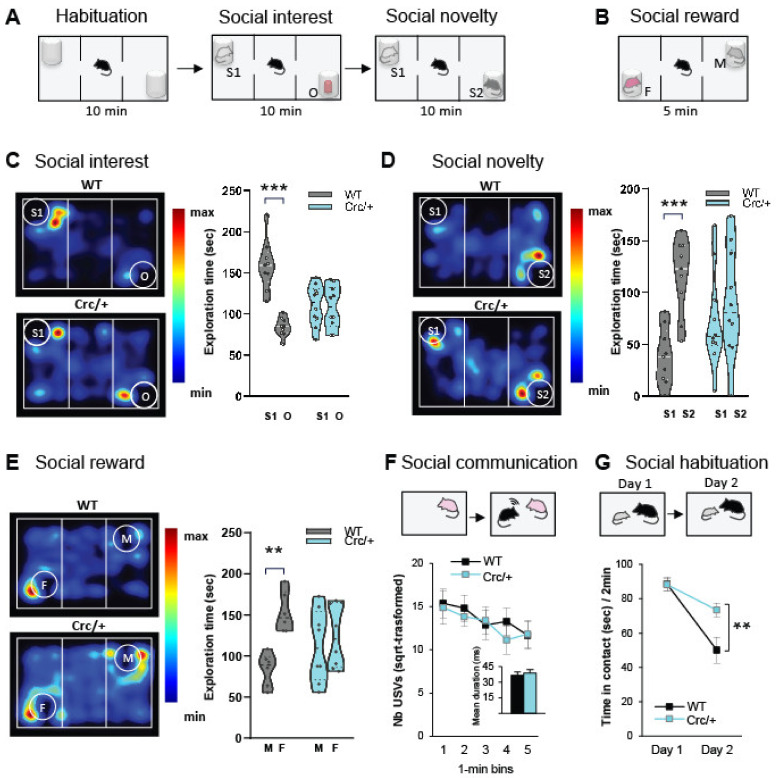
*Scrib^crc/+^* mice display selective deficits of social motivation task. (**A**,**B**) Experimental design protocol of the three-chambered test. (**C**) During the social interest test, the WT mice showed more exploration with the stranger male mice (S1) than the object (O), which was not observed in the *Scrib^crc/+^* mice (2-way ANOVA: S1 vs. O, Bonferroni corrected *t*-test: WT, *t*_16_ = 6.57, *** *p* < 0.0001; *Scrib^crc/+^*, *t*_18_ = 0.22, n.s; *n* = 9–10 mice per genotype). (**D**) During the social novelty test, the WT mice had significantly more interactions with the novel male mice (S2) than the familiar male (S1) mice (2-way ANOVA: S1 vs. S2, Bonferroni corrected *t*-test: WT: *t*_16_ = 3.84, *** *p* < 0.001; *Scrib^crc/+^*: *t*_18_ = 0.84, n.s; *n* = 9–10 mice per genotype). (**E**) During the social reward test, the WT mice showed a preference for the female mice (F) versus male (M) while the *Scrib^crc/+^* mice did not (2-way ANOVA: F vs. M; Bonferroni corrected *t*-test: WT: *t*_24_ = 3.48, ** *p* < 0.01; *Scrib^crc/+^*: *t*_24_ = 0.70, n.s; *n* = 6–8 mice per genotype). (**F**) In the direct social interaction with females, the number of ultrasonic vocalizations was not affected in the *Scrib^crc/+^* mice (2-way ANOVA: *F*_1,22_ < 1, n.s for genotype effect; *n* = 12–13 mice per genotype). (**G**) Mean investigation duration with a juvenile during initial social recognition (day 1) and social habituation (day 2). *Scrib^crc/+^* mice had a deficit in social habituation (RMANOVA: WT vs. *Scrib^crc/+^*, Bonferroni corrected *t*-test: day 2: *t*_24_ = 3.40, ** *p* < 0.01; *n* = 11–15 mice per genotype). For (**C**–**E**), data are presented as median with 25th and 75th percentile, and single data points are shown as dots. For (**F**,**G**), data are presented as mean ± SEM. The shaded area represents the probability distribution of the variable.

**Figure 2 cells-11-01601-f002:**
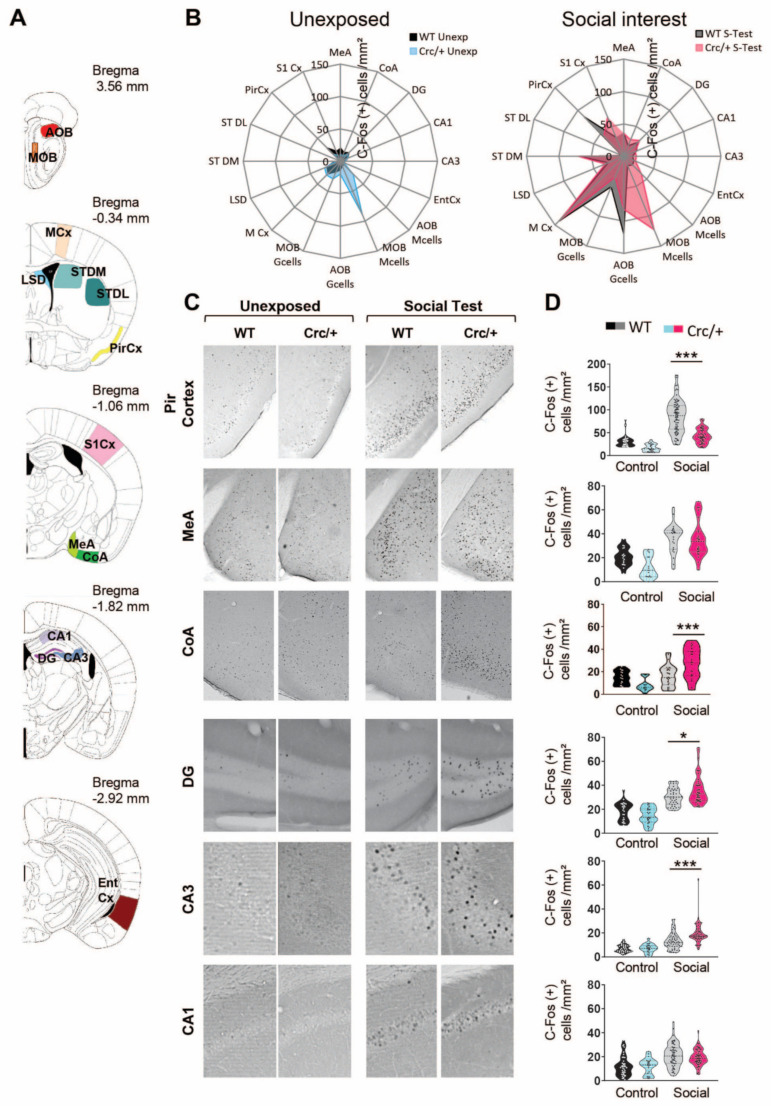
Specific patterns of neural activation after social interest test in *Scrib^crc/+^* mice. (**A**) Coronal section of neuroanatomical areas analyzed for c−Fos immunoreactivity after a social interest test adapted from Paxinos and Franklin (1997), showing the accessory olfactory bulb (AOB) and the main olfactory bulb (MOB) granular (Gcells) and mitral cell subdivisions (Mcells), as well as the motor cortex (MCx), the dorsal portion of the lateral septum (LSD), the dorsomedial (STDM) and dorsolateral (STDL) parts of the striatum, the piriform cortex (PirCx), the somatosensoriel cortex (S1Cx), the medial nucleus (MeA) and cortical nucleus of the amygdala (CoA), the CA1 (CA1) and the CA3 (CA3) subregions of the hippocampus, the dentate gyrus (DG), and the lateral entorhinal cortex (EntCx). (**B**) Radar chart recapitulating the activate brain region as measured by the change in c-Fos immunoractive cells in the WT and *Scrib^crc/+^* mice in the control and social condition. (**C**) Representative microphotographs showing c-Fos-positive cells (dark dots) in the PirCx, MeA, CoA, DG, CA3, and CA1 of WT mice or *Scrib^crc/+^* mice 1 h after the control or social test. (**D**) Number of c-Fos-positive cells per mm^2^ are presented as median with 25th and 75th percentiles, and single data points are shown as dots for each genotype and exposure condition. The shaded area represents the probability distribution of the variable. Animals per condition *n* = 3–6. Number of values per area: PirCx (*n* = 28–96), MeA (*n* = 15–25), CoA (*n* = 14–28), DG (*n* = 25–46), CA3 (*n* = 20–57), CA1 (*n* = 22–60). One-way ANOVA: * *p* ≤ 0.05 and *** *p* ≤ 0.001.

**Figure 3 cells-11-01601-f003:**
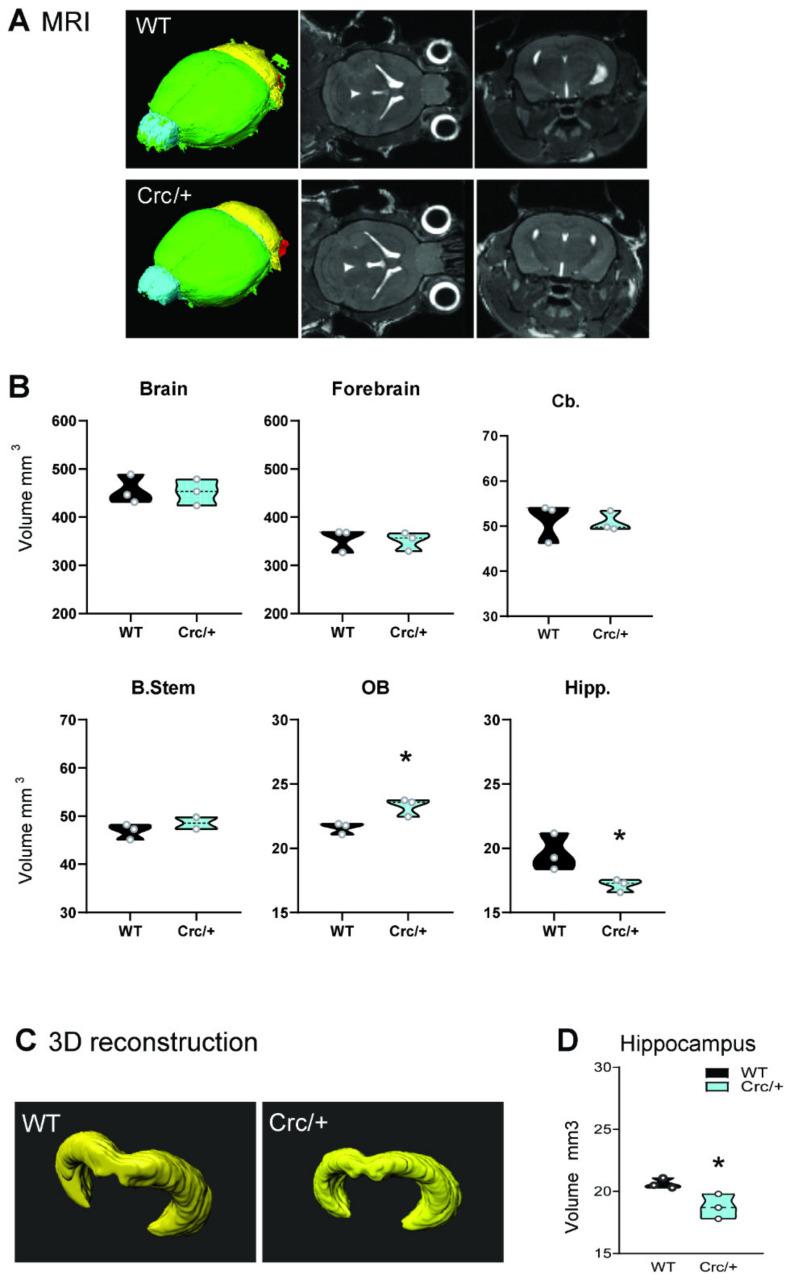
Scrib1 mutation decreasing the hippocampus volume. (**A**) MRI 3D reconstruction (left), axial (middle), and coronal (right) maps of a WT and *Scrib^crc/+^* mouse brain in vivo. (**B**) MRI volume quantification showing a specific reduction in the hippocampus volume and increased olfactory bulb volume in a *Scrib^crc/+^* mouse brain (*n* = 3 per genotype; *t*-test: * *p* ≤ 0.05). Forebrain (green area), cerebellum (Cb, yellow area), brain stem (B.Stem, red area), olfactory bulb (OB, cyan area), and hippocampus (Hipp). (**C**) Imaris 3D reconstruction of a WT and *Scrib^crc/+^* mouse hippocampus. (**D**) Volume quantification showing a reduction of hippocampal formation in the *Scrib^crc/+^* mice (*n* = 2–3 per genotype; *t*-test: * *p* ≤ 0.05). All data are presented as median with 25th and 75th percentiles, and single data points are shown as dots. The shaded area represents the probability distribution of the variable.

**Figure 4 cells-11-01601-f004:**
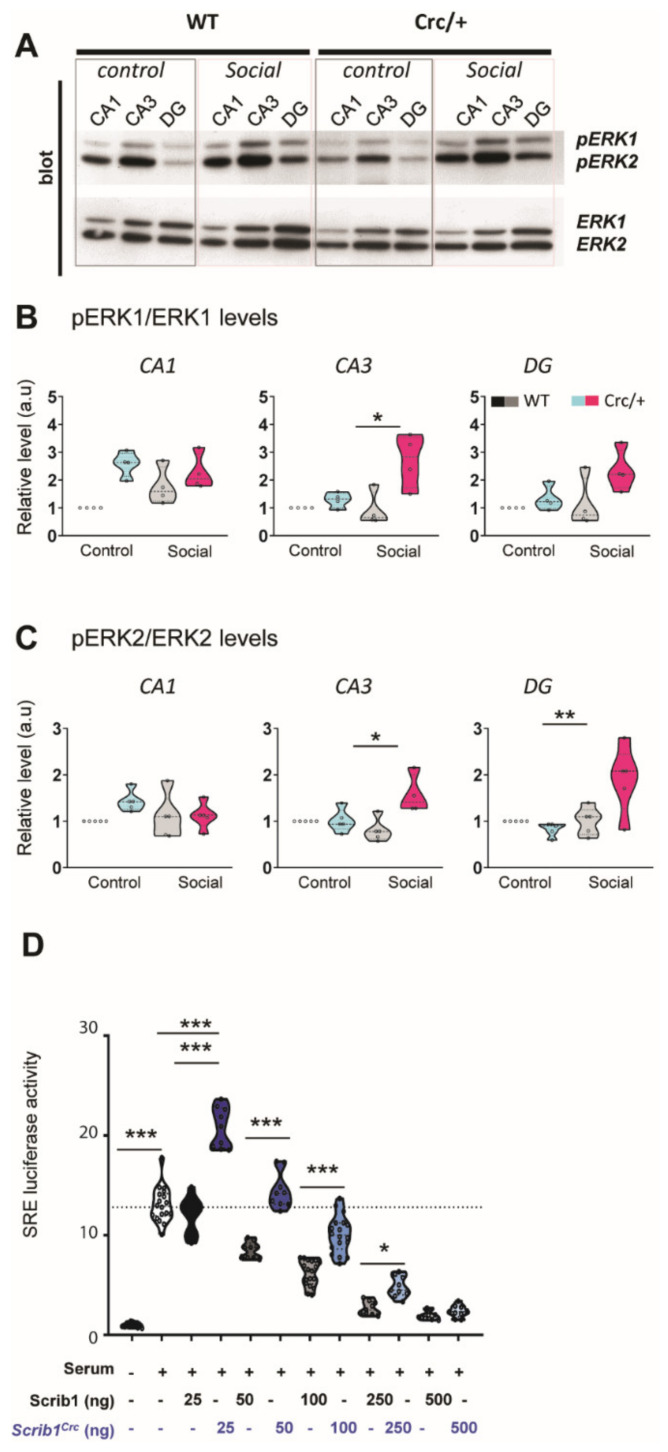
Scrib1 mutation modulating ERK protein levels in the hippocampus and having a biphasic effect on the ERK signaling pathway. (**A**) Representative immunoblot illustration of CA1, CA3, and DG from control and social WT and *Scrib^crc/+^* mice analyzed simultaneously for phospho−ERK1/2 (pERK1/2) and ERK1/2. (**B**,**C**) Quantifications of phospho−ERK1 and phospho−ERK2 in CA1, CA3, and DG (2-way ANOVA; social WT vs. social *Scrib^crc/+^*; pERK1/ERK1−CA3: *F*_1,12_ = 6.41, * *p* < 0.05; social pERK2/ERK2−DG: *F*_1,16_ = 8.96, ** *p* < 0.01; pERK2/ERK2−CA3: *F*_1,15_ = 9.88, ** *p* < 0.01 for genotype x exposure interaction; *n* = 5–8 values per conditions). (**D**) Luciferase reporter assay data from HEK293 cells transfected with the SRE-LUC reporter vector, along with the increasing amount of the Scrib or *Scrib^crc/+^* expression plasmid as indicated (2−way ANOVA: *F_11.132_* = 217.4; **** p* < 0.0001 for dose effect). Activation threshold of the serum-induced ERK pathway is indicated by the horizontal dashed line (Scrib vs. Scrib^crc^, Bonferroni comparison *t*-test: 25 ng: *t_16_* = 13.68, *** *p* < 0.001; 50 ng: *t_16_* = 9.248, *** *p* < 0.001; 100 ng: *t_34_* = 8.57, *** *p* < 0.001; 250 ng: *t_16_* = 3.31, * *p* < 0.05; 500 ng: *t_16_* = 3.31, n.s; *n* = 9 values per condition). All data are presented as median with 25th and 75th percentiles, and single data points are shown as dots. The shaded area represents the probability distribution of the variable.

**Figure 5 cells-11-01601-f005:**
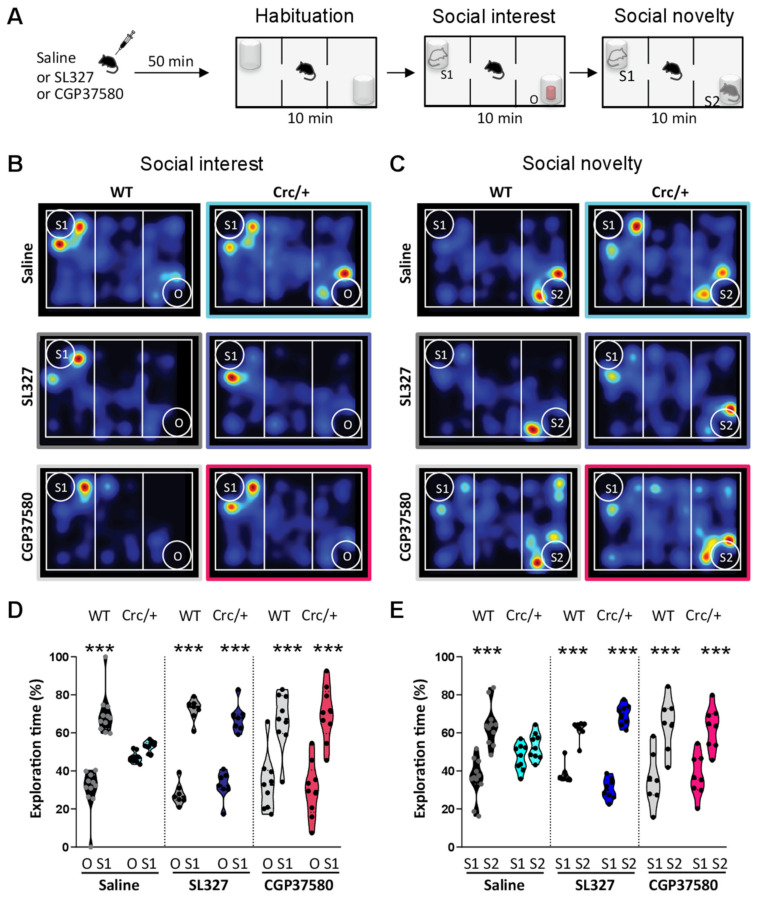
The inhibition of ERK or Mnk1 in *Scrib^crc/+^* mice reversed sociability deficits. (**A**) Experimental design protocol of the three-chambered choice test. (**B,C**) Representative heat map of social interest and social novelty behaviors. (**D**) Time spent sniffing with stranger male mice (S1) versus object (O) during the social interest test after vehicle, SL327, or CGP37580 treatment (2–way ANOVA: O vs. S1; Bonferonni’s multiple comparaisons test: WT–Saline, *t_11_* = 8.80, *** *p* < 0.0001; *Scrib^crc/+^*–*saline*, *t_9_* = 1.17, n.s; WT–SL327, *t_7_* = 7.87, *** *p* < 0.0001; *Scrib^crc/+^*–*SL327*, *t_7_* = 6.35, *** *p* < 0.0001; WT–CGP57380, *t_9_* = 6.70, *** *p* < 0.0001; *Scrib^crc/+^*–CGP57380, *t_9_* = 8.04, *** *p* < 0.0001; *n* = 7–11 animals per condition). (**E**) Time spent sniffing with novel male mice (S2) versus familiar male mice (S1) during the social novelty test after vehicle, SL327, or CGP37580 treatment (2–way ANOVA: S1 vs. S2; Bonferonni’s multiple comparaisons test: WT–Saline, *t_11_* = 5,81, *** *p* < 0.0001; *Scrib^crc/+^*–*saline*, *t_9_* = 1.09, n.s; WT–SL327, *t_8_* = 4.20, *** *p* < 0.001; *Scrib^crc/+^*–SL327, *t_8_* = 6.90, *** *p* < 0.001; WT–CGP57380, *t_9_* = 4;46, *** *p* < 0.0001; *Scrib^crc/+^*–CGP57380, *t_8_* = 4.46, *** *p* < 0.0001; *n* = 7–11 animals per condition). All data are presented as median with 25th and 75th percentiles, and single data points are shown as dots. The shaded area represents the probability distribution of the variable.

## Data Availability

The published article includes all statistical analysis and all full blots analyzed during this study. All the others originals data for figures supporting the current study are available from the correspondence authors.

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
