# Peer review of "Scribble Controls Social Motivation Behavior through the Regulation of the ERK/Mnk1 Pathway"

_cells, 2022, doi:10.3390/cells11101601_

Round 1
Reviewer 1 Report
The article suggests the interactions promoted by the Scrib gene related to the hippocampus concerning ASD. Understanding how Scrib is an important regulator of selected social behavior and the biochemical signaling path involved, contributes to understanding the pathophysiology of ASD. I suggest some changes to add to the text:
1) Add an experimental timeline
2) What strain is used for knockout animals?
3) Approval number of the ethics committee for using animals in research, where the animals were kept.
4) Why was it analyzed in adult animals? Please describe briefly in the thread
5)Add reference to the methodology (items 2.5, 2.7, 2.8,2.9,2.11.2.12,2.13,2.16).
6) Include antibody dilution in the methodology.
7)Add the images of all blotting membranes to the supplemental material.
Reviewer 2 Report
Moreau et al., delve into the analysis of altered social behavior associated with developmental pathologies such as ASD. They use a Scribcrc/+ heterozygous transgenic animal model, with a mutation in the SCRIB polarity gene which encodes for the Scrib protein which modulates MAPK signaling. Accumulating evidences suggest SCRIB, together with other polarity genes, has a main role in neurodevelopmental disorders. Through an extensive battery of behavioral tests to establish the relationship between the SCRIB gene and ASD-like behaviors, as well as pharmacological treatments, cell culture and MRI, Moreau et al. demonstrate how these mice show alterations in social behavior, associated with an over-activation of hippocampal phosphorylated ERK and increased c-fos activity (more evident in DG and CA3 regions). These behavioral alterations can be reversed by acting on ERK phosphorylation directly (MEK inhibition) or indirectly (Mnk1 inhibition). The article by Moureau et al. is exquisitely presented, its methodology is impeccable and its conclusions are in agreement with the results obtained. In terms of applicability, it postulates the MAPK/ERK pathway as a possible therapeutic target to address social recognition problems in both ASD-like and NTD.
The references are very adequate and the number of self-citations is scarce and fully justified.
Despite the high quality of the manuscript there are a number of issues to consider.
- Line 404. The authors indicate that in WT mice there is an increase in c-Fos-positive cells compared to the control condition in MOB Gcells. However, the Supplemmental Figure 4A/B shows an increase in the accessory and not in the main olfactory bulb .
- Line 410. It is indicated that the number of c-Fos-positive cells was significantly higher in the olfactory bulb (AOB/MOB), however this increase is only observed in mitral cells and not in granular cells.
- Line 429. The CoA results are very significant and show the most robust increase. Authors are suggested to include CoA graph in Figure 2 and not in Supplemental Figure 4.
- Line 432. The figure 2D doesn’t exist.
- Figure 3A: looks like an axial slice of the brain and not sagittal one.
- Figure 3B (Legend) replace "brain strem" by "brain stem".
- Figure 3B (Legend) replace "olfacory bulb" by "olfactory bulb".
- Figure 3B (Brain stem): there are only two specimens represented in Scribcrc/+ group
- Figure 3: X-axis legends of B. Stem, OB and Hipp. graphs are missing.
Finally, I would like to thank the authors for the high methodological quality and special scientific relevance of their manuscript.
Author Response
"Please see the attachment."

Reviewer 3 Report
This work identifies Scrib as an important regulator of selected social behaviors in mice.
Importantly, data indicate positive modulation of phenotype through negative regulation of the ERK/Mnk signaling pathway.
Finally, the circletail mutant mouse is presented as a useful model on which the effects on social preference/recognition of candidate therapeutical compounds could be tested.
Author Response
We thank the reviewer for the nice comments.
